# Development of Genetic Tools in Glucoamylase-Hyperproducing Industrial *Aspergillus niger* Strains

**DOI:** 10.3390/biology11101396

**Published:** 2022-09-24

**Authors:** Dandan Liu, Qian Liu, Wenzhu Guo, Yin Liu, Min Wu, Yongli Zhang, Jingen Li, Wenliang Sun, Xingji Wang, Qun He, Chaoguang Tian

**Affiliations:** 1State Key Laboratory of Agrobiotechnology and MOA Key Laboratory of Soil Microbiology, College of Biological Sciences, China Agricultural University, Beijing 100193, China; 2Key Laboratory of Systems Microbial Biotechnology, Tianjin Institute of Industrial Biotechnology, Chinese Academy of Sciences, Tianjin 300308, China; 3National Technology Innovation Center of Synthetic Biology, Tianjin 300308, China; 4Longda Biotechnology Inc., Linyi 276400, China

**Keywords:** glucoamylase, *Aspergillus niger*, protoplast, *Agrobacterium tumefaciens*, genetic tool, CRISPR/Cas9, marker-free engineering, transformation efficiency

## Abstract

**Simple Summary:**

Glucoamylase is one of the most needed industrial enzymes in the food and biofuel industries. *Aspergillus niger* is a commonly used cell factory for the production of commercial glucoamylase. For decades, genetic manipulation has promoted significant progress in industrial fungi for strain engineering and in obtaining deep insights into their genetic features. However, genetic engineering is more laborious in the glucoamylase-producing industrial strains *A. niger* N1 and O1 because their fungal features of having few conidia (N1) or of being aconidial (O1) make them difficult to perform transformation on. In this study, we targeted *A. niger* N1 and O1 and successfully developed high-efficiency transformation tools. We also constructed a clustered regularly interspaced short palindromic repeat (CRISPR)/Cas9 editing marker-free system using an autonomously replicating plasmid to express Cas9 protein and to guide RNA and the selectable marker. By using the genetic tools developed here, we generated nine albino deletion mutants. After three rounds of sub-culturing under nonselective conditions, the albino deletions lost the autonomously replicating plasmid. Together, the tools and optimization process above provided a good reference to manipulate the tough working industrial strain, not only for the further engineering these two glucoamylase-hyperproducing strains, but also for other industrial strains.

**Abstract:**

The filamentous fungus *Aspergillus niger* is widely exploited by the fermentation industry for the production of enzymes, particularly glucoamylase. Although a variety of genetic techniques have been successfully used in wild-type *A. niger*, the transformation of industrially used strains with few conidia (e.g., *A. niger* N1) or that are even aconidial (e.g., *A. niger* O1) remains laborious. Herein, we developed genetic tools, including the protoplast-mediated transformation and *Agrobacterium tumefaciens*-mediated transformation of the *A. niger* strains N1 and O1 using green fluorescent protein as a reporter marker. Following the optimization of various factors for protoplast release from mycelium, the protoplast-mediated transformation efficiency reached 89.3% (25/28) for N1 and 82.1% (32/39) for O1. The *A. tumefaciens*-mediated transformation efficiency was 98.2% (55/56) for N1 and 43.8% (28/64) for O1. We also developed a marker-free CRISPR/Cas9 genome editing system using an AMA1-based plasmid to express the Cas9 protein and sgRNA. Out of 22 transformants, 9 *albA* deletion mutants were constructed in the *A. niger* N1 background using the protoplast-mediated transformation method and the marker-free CRISPR/Cas9 system developed here. The genome editing methods improved here will accelerate the elucidation of the mechanism of glucoamylase hyperproduction in these industrial fungi and will contribute to the use of efficient targeted mutation in other industrial strains of *A. niger*.

## 1. Introduction

Glucoamylases (1,4-α-D-glucan glucohydrolases; EC 3.2.1.3) are mainly used to hydrolyze α-1,4 glycosidic bonds at the non-reducing end of starch to cut off the glucose unit [1,2]. Fungal glucoamylases are among the most widely used enzyme preparations in industry, notably in the food industry, including in the production of glucose syrup, fructose syrup, and ethanol [3,4,5,6]. The product portfolio of filamentous fungi is undoubtedly extensive. They are widely used in the production of recombinant proteins and enzymes [7,8]. Notably, *Aspergillus niger*, which has Generally Recognized as Safe status, is harnessed as a cell factory for the production of a diverse range of enzymes, including glucoamylase, cellulase, xylanase, amylase, protease, and lipase [9,10,11,12,13,14]. *A. niger* CBS 513.88 is a model strain, and its whole genome has been sequenced [15]. Strain CBS 513.88 is considered an ancestor of the enzymes that are currently used as production strains, and various studies have been performed on the strain CBS 513.88, such as for the improvement of glucoamylase production [13,16].

An efficient transformation system is imperative for the genetic manipulation of filamentous fungi [17]. Similar to other model filamentous fungi such as *Neurospora crassa* and *Aspergillus nidulans*, genetic transformation methods have been developed for *A. niger*, including *Agrobacterium tumefaciens*-mediated transformation (AMT), protoplast-mediated transformation (PMT), electroporation, and particle bombardment [18]. *A. tumefaciens* is a plant pathogen that is capable of causing crown gall tumors on plants by transferring a part of its DNA (T-DNA) that is located on a tumor-inducing (Ti) plasmid through a type IV secretion system to the host [19]. Meyer et al. reviewed the advantages and disadvantages of these transformation methods [20]. AMT and PMT are the most commonly used methods and have been successfully applied in the transformation of many filamentous fungi, including *A. niger*, with variable efficiencies [18,21,22]. Bundock et al. demonstrated that AMT was based on the ability of *A. tumefaciens* cells to transfer part of their T-DNA; this method is commonly used in plants and yeast [23]. Then, AMT was used in the genetic transformation of several filamentous fungi, including *A. awamori*; one study found that DNA could be transferred between the prokaryote and the fungus, and, in most transformants, a single copy of the T-DNA was randomly integrated into the genome of the fungus [24]. Since then, the AMT method has been used in a great variety of filamentous fungi [21,25,26,27,28,29].

A protoplast is a naked cell in which the cell wall has usually been removed using lysing enzymes; it can be applied in cell fusion or transformation. After pioneering fungal protoplast isolation and transformation in *Saccharomyces cerevisiae*, protoplast isolation and transformation based on protoplasts have been used for filamentous fungi, including in *A. niger* strains [18,30,31]. AMT does not require the excessive treatment of fungi in the process of transformation; thus conidia, germinated conidia, and vegetative mycelia can all be used for the transformation [25,32]. AMT also has high transformation efficiency and produces stable transformants. However, compared to PMT, AMT is complicated and time consuming. PMT might also be the better method when multiple copies of the expressed gene need to be integrated into the genome [33,34].

Notwithstanding that the genetic manipulation of wild-type *A. niger* has been established [20,25], many industrially used strains of *A. niger* are still difficult to transform because the industrial strains have undergone long-term mutagenesis. Compared to the wild-type varieties, the cell structure of these industrial strains has undergone great changes, such as losing most of its conidiation capacity, and even the conidia coats are significantly different from those of the wild-type varieties [35]. *A. niger* N1 and O1 are glucoamylase-hyperproducing industrial strains derived from the same original strain. *A. niger* strain O1 produces no conidia, whereas *A. niger* strain N1 produces some conidia. The glucoamylase components and production levels are also different between these strains.

Precision genome editing by clustered regularly interspaced short palindromic repeat (CRISPR)-associated RNA-guided DNA endonucleases (Cas9) has rapidly become a widely used technology. Functional CRISPR/Cas9 systems for gene editing have been successfully developed in filamentous fungi, including in the industrial filamentous fungi *Aspergillus* spp. [36,37], *Trichoderma reesei* [38], and *Penicillium chrysogenum* [39]. In previous studies, Cas9 was either integrated into the genome [37,38,40] or transiently expressed from a nonreplicating plasmid introduced into protoplasts [41,42,43]. Cas9 can also be expressed from plasmids [36] containing autonomous maintenance in the *Aspergillus* (AMA1) sequence or from the autonomously replicating sequences (ARSs) in *Ustilago maydis*. Additionally, some studies have reported the use of marker-free deletion for single or multiple genes in one transformation using repair DNA fragment(s) in combination with Cas9-expressing plasmids with self-replicating extrachromosomal AMA1 elements [44,45,46,47,48]. However, gene editing without the use of integrative selection markers has rarely been reported in glucoamylase-hyperproducing industrial *A. niger* strains.

In previous studies of PMT, 10^8^ conidia from *Aspergillus giganteus* strain IfGB 15/0903 and *Myceliophthora thermophila* wild-type strain ATCC 42,464 were incubated in liquid culture to generate a sufficient yield of protoplasts, with 55 and 25 transformants being obtained in *A. giganteus* and *M. thermophila*, respectively [49,50]. The AMT system was constructed in *M. thermophila* ATCC 42,464, and the transformation efficiency was about 58% for a pool of 116 transformants analyzed within the time period of 19–21 d [21]. In *Aspergillus awamori* strain CBS115.52, using the AMT system, hygromycin-resistant transformants were obtained at a frequency of 200–250 transformants per 10^6^ conidiospores within 10–12 d [19]. Although the AMT system is usually successful in filamentous fungi, there are also some cases in which AMT is less successful or even fails to produce transformants, e.g., in *A. niger* and *Sclerotinia sclerotiorum* (reviewed by Michielse et al. [25]). Moreover, it is very difficult to obtain abundant 10^8^ conidia per mL in glucoamylase-hyperproducing industrial strains with small colonies and few conidia or even in strains that are aconidial. Therefore, it is imperative to develop a genetic transformation method suitable for industrial filamentous fungi, especially those with poor growth or in aconidial industrial strains.

In this study, we targeted the glucoamylase-producing industrial strains *A. niger* N1 and O1. We successfully developed high-efficiency transformation tools via the systematic optimization of the whole process. To the authors’ knowledge, the PMT method has not been reported for filamentous fungi that do not produce conidia. Thus, we herein successfully established PMT for the non-conidial strain *A. niger* O1 by using a culture of strain O1 on agar plates to obtain young mycelia for the release of protoplasts, an approach that has not been studied previously. The final transformation efficiency reached 89.3% for *A. niger* N1 and 82.1% for *A. niger* O1 when using PMT. The final transformation efficiency obtained using AMT was 98.2% for *A. niger* N1 and 43.8% for *A. niger* O1. Using PMT and the marker-free CRISPR/Cas9 system developed in this study, an *albA* deletion strain was constructed in the *A. niger* N1 background. The optimized processes developed here provide a reference for the manipulation of industrial strains, not only for the further engineering of these two glucoamylase-producing strains of *A. niger*, but also for other industrial *Aspergillus* strains and other filamentous fungi.

## 2. Materials and Methods

### 2.1. Strains, Media, Culture Conditions, and Plasmids

The *A. niger* N1 and O1 used in this study were kindly provided by Longda Biotechnology (Shandong, China). As a wild-type control strain, *A. niger* CBS 513.88 was cultured on potato dextrose agar (PDA) to display the normal production of conidia compared to the industrial strains *A. niger* N1 and O1, which produced few conidia or no conidia. *A. niger* N1 was cultured on Czapek–Dox new (CPZ-new) agar in 9 cm Petri dishes or in eggplant-shaped culture flasks for 7-10 days to produce conidia. The composition of the CPZ-new agar was: 21.8% sorbitol, 0.2% KNO_3_, 0.38% NaH_2_PO_4_, 0.13% MgSO_4_, 0.13% KCl, 0.04% Na_2_B_4_O_7_, 0.08% Na_2_MoO_4_, 0.04% CuSO_4_, 0.12% FeSO_4_•7H_2_O, 0.01% MnSO_4_•H_2_O, 0.1% ZnSO_4_•7H_2_O, 5 mL glycerol, and 7.5 g agar. It was adjusted to pH 6.0, made up to 500 mL with double-distilled H_2_O, and autoclaved. A. *niger* O1 was cultured on Czapek–Dox old (CPZ-old) agar in 9 cm Petri dishes; the medium composition was the same as described previously [51]. Lysing enzymes from *T. harzianum* (Sigma, St. Louis, USA), Lysozyme (Solarbio, Beijing, China), Snailase (LABLEAD, Beijing, China), Driselase (Sigma), and Yatalase (Takara, Nishinomiya, Japan) were used for protoplast isolation. The MM salts (2.5×; 1 L) used for the induction medium (IM) contained 3.625 g K_2_HPO_4_, 5.125 g KH_2_PO_4_, 1.25 g MgSO_4_•7H_2_O, 0.375 g NaCl, 0.0062 g FeSO_4_•7H_2_O, and 1.25g (NH_4_)_2_SO_4_. The M-100 trace element solution (0.5 L) contained 30 mg H_3_BO_3_, 70 mg MnCl•4H_2_O, 200 mg ZnCl_2_, 20 mg NaMoO_4_•2H_2_O, 50 mg FeCl_3_•6H_2_O, and 200 mg CuSO_4_•5H_2_O. The M-100 salt solution (1 L) contained 16 g K_2_HPO_4_, 4 g NaSO_4_, 8 g KCl, 2 g MgSO_4_•7H_2_O, 1 g CaCl_2_, and 8 mL M-100 trace element solution. IM (200 mL) contained 80 mL 2.5× MM salts, 0.36 g 10 mM glucose, 5‰ glycerol, 40 mM MES, and 200 µM acetosyringone (AS). The M-100-agar (1 L) contained 62.5 mL M-100 salt solution, 10 g glucose, 3 g KNO_3_, and 1.5% agar.

The binary vector pAN52-P*ahr-gfp*-T*ahr* was constructed for the PMT of *A. niger* N1 and O1. The pAN52-P*ahr*-*gfp*-T*ahr* vector contained the P*ahr* promoter of the alkyl hydroperoxide reductase from *M. thermophila*, enhanced green fluorescent protein (GFP), the promoter (P*trpC*) of the tryptophan synthetase gene from *A. nidulans*, and the neomycin resistance gene (*neo*). The *gfp*–*neo* cassette fragment was amplified from pAN52-P*ahr*-*gfp*-T*ahr* using the primers 1F/1R (Appendix A) and was also used for the transformation of *A. niger* N1 and O1 into protoplasts. A Ti vector, pPK2-*hph*-*gfp,* containing the P*trpC* promoter from *A. nidulans* and the *tef1* promoter (P*tef1*) from *A. niger* was developed as the test plasmid for the AMT of *A. niger* N1 and O1. The plasmid pPK2-*hph*-*gfp* contains a GFP reporter marker as well as a hygromycin resistance gene (*hph*). The skeleton of pPK2-*hph*-*gfp* was reported in a previous study [21].

For the conidia counting analysis, *A. niger* N1 and *A. niger* CBS513.88 were cultured on CPZ-new agar and PDA in 9 cm Petri dishes to produce conidia, respectively. Then, 10 mL of sterile normal saline was added to each plate to harvest the conidia by gently scraping the agar with a sterile stick to prepare a suspension. The resulting suspension was filtered through sterile two-layer lens paper to remove mycelial debris, and the conidia were centrifuged at 4000 rpm for 10 min. The pellet was resuspended gently in 1 mL sterile normal saline, and 10 µL of the conidia solution was quantified using a hemocytometer. For morphological analysis, conidia from the *A. niger* strain N1 were cultivated on several commonly used media for filamentous fungi, including CPZ-old, MM1 (Vogel’s minimal medium) [52], MM2 (complete medium) [14], PDA, and EP (EP complete medium) [53] at 30 °C for 7 days. For marker screening, *A. niger* N1 and O1 were cultivated on bottom-agar plates at 30 and 34 °C, respectively [54].

### 2.2. Preparation and Transformation of A. niger N1 and O1 Mediated by Protoplasts

PMT was performed on *A. niger* by modifying a method previously described for *A. nidulans* [55]. *A. niger* N1 conidia were cultivated in an eggplant-shaped culture flask in CPZ new medium for 7-10 days, and the conidia were collected. The culture conditions used to produce mycelia for protoplast isolation were specific to each strain (*A. niger* N1 and O1), so mycelia were cultured in five different media to investigate the optimal conditions. For *A. niger* strain N1, a conidial suspension (10^7^/mL) was prepared with normal saline, and 100 µL of that suspension was coated on a 9 cm Petri dish containing CPZ-old, MM1, MM2, PDA, or EP medium, covered with a piece of cellophane, and incubated at 30 °C. Then, mycelium was harvested at different times to test the effect of mycelial age on protoplast yield. For *A. niger* O1, which does not produce conidia, using a stick, we scraped young mycelia from CPZ-old agar onto PDA, MM1, MM2, or EP in a 9 cm Petri dish covered with a piece of cellophane. Unlike in the culturing method for *A. niger* mycelia using liquid medium [14] for the release of protoplasts, herein, the young mycelia of *A. niger* O1 were scraped onto the solid plate. Ten sheets of cellophane with mycelia attached were completely immersed in a 9 cm Petri dish containing 25 mL of enzymatic hydrolysate dissolving solution A (0.1 M KH_2_PO_4_ and 1.2 M sorbitol, pH adjusted to 5.6 with KOH) at 30 °C. Unlike methods that use inorganic KCl as the osmotic stabilizer [55], we used organic sorbitol. The lytic enzyme preparation Novozyme 234 is not currently available for purchase, so we applied lysing enzymes from *T. harzianum*, Lysozyme, Snailase, Driselase, and Yatalase. Protoplast cleavage was observed under a microscope every half hour. The effects of enzymatic incubation for 0.5, 1, 2, 3, 4, and 5 h were also determined. The resulting suspension was filtered through sterile two-layer lens paper to remove mycelial debris, and the protoplasts were centrifuged at 3000 rpm for 10 min.

For the PMT of A. *niger* N1 and O1, the supernatant from the above centrifugation was discarded, and the pellet was washed with 5–10 mL of solution B (50 mM CaCl_2_, 1 M sorbitol, 10 mM Tris-Cl (pH 7.5), pH adjusted to 5.6 with HCl) and then centrifuged at 3000 rpm for 10 min. The pellet was resuspended gently in 220 µL of solution B, and 10 µL of the protoplast solution was counted using a hemocytometer. Linear *gfp*–*neo* cassette DNA (10 µg) was added and mixed gently. Next, 70 µL of PEG solution (25% PEG 6000 and 50 mM CaCl_2_ in 10 mM pH 7.5 Tris-Cl) was added and mixed gently, and the mixture was incubated on ice for 20 min. PEG solution (2 mL) was added to the tube dropwise, and the tube was gently swirled before being left at room temperature for 5 min. Then, 4 mL of solution B was added, and the tube was gently swirled. Medium (containing 2% sucrose, 2 mL 50 × Vogel’s salts, 1 M sorbitol, 1.5% agarose, and 200–300 µg/mL G418 in 100 mL) was added to the above mixture to a volume of 50 mL. The tube was gently inverted, and the mixture was poured over seven bottom media (containing 2% sucrose, 2 mL 50 × Vogel’s salts, 1 M sorbitol, 1.5% agar, and 200–300 µg/mL G418 in 100 mL) plates. Transformants were selected after culturing at 30 °C (*A. niger* N1) or at 34 °C (A. *niger* O1) for 4–7 days.

### 2.3. Preparation and Transformation of A. niger N1 and O1 Mediated by A. tumefaciens

*Agrobacterium*-mediated genetic transformation was performed by modifying a method previously described for *M. thermophila* [21]. First, a single colony of *A. tumefaciens* harboring pPK2-*hph*-*gfp* was inoculated into Luria–Bertani medium supplemented with 50 μg/mL of kanamycin and was cultivated at 220 rpm for 12–20 h at 28 °C to an optical density of 0.5–1.0 at 600 nm (OD600). Xu et al. reported that *A. tumefaciens* needs to be washed twice with IM containing AS [21]. However, herein, the transformants were obtained without the *A. tumefaciens* being washed. Then, the cells were harvested by centrifugation at 4000 rpm for 10 min and were directly diluted with IM containing 200 μM AS and 10 mM MES to OD600 = 0.1–0.2. The culture was grown at 28 °C with shaking at 220 rpm for 6–8 h; it could then be used for cocultures for the transformation of *A. niger* (see below).

Conidia of *A. niger* N1 or young mycelia of *A. niger* O1 were prepared as described above (see Section 2.1). *A. niger* N1 conidia were cultivated in eggplant-shaped culture flasks containing CPZ-new agar for 7–10 days. Sterile normal saline was added to the flasks to harvest the conidia by gently scraping the agar with a sterile stick to prepare a conidial suspension (10^6^/mL). Using a sterile stick, we scraped young mycelia of *A. niger* O1 grown on CPZ-old agar for 2 days onto fresh CPZ-old agar covered with a piece of cellophane in a 9 cm Petri dish and incubated the dish for 19 h at 34 °C. In contrast to a previous study [21], the young mycelia of *A. niger* O1, but not the conidia, were co-incubated with *A. tumefaciens*, which provided a reference for the AMT of non-conidial filamentous fungi.

Finally, *A. tumefaciens* and *A. niger* were cocultured. The *A. tumefaciens* culture (100 µL) was mixed with an equal volume of *A. niger* N1 conidia (10^6^/mL) or young mycelia of *A. niger* O1 and then spread onto an IM agar plate (9.0 cm; supplemented with 200 µM AS and 10 mM MES) that had been covered with cellophane beforehand. After cocultivation at 28 °C for 2 days, the cellophane was transferred onto an M-100 medium plate (9.0 cm) supplemented with 50 μg/mL hygromycin (Roche, Basel, Switzerland) for *A. niger* N1 or with 200 μg/mL hygromycin for *A. niger* O1 and 300 μg/mL cefotaxime (Sigma-Aldrich, St. Louis, USA) and overlaid with M-100 medium supplemented with 50 μg/mL hygromycin for *A. niger* N1 or 200 μg/mL hygromycin for *A. niger* O1 and 300 μg/mL cefotaxime before being cultivated at 30 °C (for *A. niger* N1) or 34 °C (for *A. niger* O1) for 4–7 days.

### 2.4. Screening of Transformants by PCR, Fluorescence Stability, RT-qPCR, and Southern Blotting

First, all of the transformants were checked by PCR analysis. The genomic DNAs of the transformants were extracted as previously described [56] and were used as templates for PCR. PCR-based detection systems were designed: primer pairs GFP-1F/1R and NEO-1F/1R were used to amplify the *gfp* and the *neo* resistance genes from the genomic DNAs of the fungi transformed with the *gfp*–*neo* cassette, respectively. Only in the case of correct transformation could the 720-bp (GFP-1F/1R) and 759-bp (NEO-1F/1R) fragments be amplified. There should be no amplification of the target genes from the untransformed genomic DNA Of *A. niger* N1 or O1 (Appendix A). Similarly, GFP-2F/2R and HPH-1F/1R were used to amplify the *gfp* and *hph* from the genomic DNA of fungi transformed with pPK2-*hph*-*gfp*, respectively. The expected amplified fragments for GFP-2F/2R and HPH-1F/1R were 1253- and 1201-bp long, respectively. Genomic DNA of untransformed *A. niger* N1 or O1 was used as a negative control (Appendix A). Then, the PCR-confirmed transformants were observed by fluorescence microscopy. To verify the stability of the fluorescence, 10 PCR- and fluorescence-confirmed transformants were randomly selected, passaged twice on CPZ-new or CPZ-old agar containing antibiotics, and the fluorescence signals were observed again.

To determine the copy number(s) of the integrated *gfp* gene in transformants, fungal genomic DNAs were used as templates for RT-qPCR using a previously described method [54]. qPCR was performed with SYBR Green Real-time PCR Master Mix (TOYOBO, Osaka, Japan) using the CFX96 Real-Time PCR Detection System (Bio-Rad, Hercules, CA, USA). The actin gene (ANI_1_106134) was used as an internal control. Each 20 μL reaction mixture contained 1 μL of diluted DNA (50–100 ng/μL), 10 μL of 2× SYBR Green Real-time PCR Master Mix, 0.8 μL of each primer (0.4 μM), and 7.4 μL of H_2_O. qPCR was performed as follows: 95 °C for 30 s and 40 cycles of 95 °C for 30 s; 55 °C for 30 s; and 72 °C for 30 s. To determine the amplification efficiencies of all of the reactions, genomic DNA samples were diluted serially to construct standard curves and were then subjected to RT-qPCR three times. All of the primers used are listed in Appendix A.

Although PCR reactions allow the recognition of T-DNA integration events, they cannot determine if the transformed DNA is integrated into a random place in the genome or into the copy number of the T-DNA. It is therefore imperative that the transformants be further analyzed by Southern blotting. Southern blotting was performed with 20 μg of genomic DNA that was digested by *Xho*I for PMT and by *EcoR*V for AMT. Genomic DNA was extracted as previously described [54]. The digested DNA was separated by agarose gel electrophoresis, and DNA transfer was performed as previously described [57]. The *gfp* fragment was used as a template, and the 310-bp PCR-amplified product used as the probe was generated with the primers Probe-F/Probe-R (Appendix A). Probe preparation, membrane hybridization, and visualization were performed according to the manufacturer’s instructions for the DIG High Prime DNA Labeling and Detection Starter Kit II (Roche, Mannheim, Germany).

### 2.5. CRISPR/Cas9 Plasmid Design

To express Cas9 and a guide RNA from the same autonomously replicating vector using hygromycin as a selection marker for fungal transformation, plasmid pFC332 was used [36]. pFC332-*AnalbA* contains a unique *Pac*I site that was used to insert a single guide RNA (sgRNA) expression cassette based on the native *A. niger* N1 U6 promoter. To select specific sgRNA targeting in *A. niger albA* (*AnalbA*), the sgRNA target site (*AnalbA* Target) in the genome of *A. niger* N1 was identified using the sgRNACas9 tool [58]. *A. niger* N1 U6p-*AnalbA* Target-sgRNA was synthesized and was inserted into pFC332 digested with *Pac*I to generate pFC332-*AnalbA*. The 5′- and 3′- flanking fragments of *AnalbA* were separately amplified from *A. niger* N1 genomic DNA via PCR with the primer pairs *AnalbA*-5F/R and *AnalbA*-3F/R, respectively (Appendix A). The amplified 5′- and 3′- fragments were assembled and ligated into pUC118 (digested with *EcoR*I and *Xba*I) using an NEB Gibson Assembly Kit to generate a pUC118-donor DNA sequence. The donor DNA sequences amplified with the primers *AnalbA*-5F and *AnalbA*-3R were used for transformation (Appendix A).

The plasmid pFC332-*AnalbA* and the donor DNA fragment were transformed into the protoplasts of *A. niger* N1. We used 10 µg of pFC332-*AnalbA* with approximately 10 µg of donor DNA fragments for each transformation. Transformants were identified using primers *AnalbA*-DE-F/R; amplification from the genomic DNA of *A. niger* N1 was expected to generate a PCR fragment of 856 bp, whereas amplification from the *albA*-knockout strains was expected to generate a PCR fragment of 560 bp. *AlbA* deletion caused the *A. niger* colonies to be white. White colonies were streaked on CPZ-new agar containing hygromycin (50 µg/mL) to ensure that the spores harbored pFC332-*AnalbA* and were thus more likely to be transformed. Next, a single colony was picked and transferred to non-selective CPZ-new medium to allow for loss of pFC332-*AnalbA*. Finally, a streak of a single colony on both CPZ-new medium containing hygromycin (50 µg/mL) and CPZ-new medium without hygromycin was used as a control for plasmid loss. DNA was isolated from strains that had lost the pFC332-*AnalbA* plasmid, and then a hygromycin gene fragment was determined by PCR using the primers HPH-DE-F/R (1006 bp) (Appendix A).

## 3. Results

### 3.1. Growth Phenotypes of A. niger N1 and O1

The morphology of the glucoamylase-producing strains *A. niger* N1, O1, and CBS 513.88 was checked on the medium-covered plates after 7 days of growth (Figure 1A). *A*. *niger* N1 and O1 formed smaller colonies than strain CBS 513.88. The conidia concentration of *A. niger* CBS 513.88 was about 44.77 × 10^6^/mL for each plate, while *A. niger* N1 had far less conidia production (6.67 × 10^6^/mL) (Figure 1A and Appendix A). Although the mycelia of *A. niger* N1 grew well on various commonly used media, conidia could only be produced on CPZ-new medium (Figure 1B). Moreover, *A. niger* O1 did not produce conidia at all (Figure 1A). The carbon source in the CPZ-old and MM1 media was sucrose; the carbon source in MM2 was glucose; there were two carbon sources (glucose and maltose) in EP medium; PDA contains a variety of carbon sources, while the major carbon source in CPZ-new medium is a high concentration of sorbitol. Previous studies have reported that the inhibition of mycelial growth was partially recovered by supplementing low or medium concentrations of sorbitol [59]. Additionally, sorbitol can stimulate the germination and growth of some xerophilic fungi [60]. Here, we found that when used as the major carbon source, sorbitol promoted the production of conidia by *A. niger* strain N1.

### 3.2. Marker Screening of A. niger N1 and O1

To improve the genetic transformation methods for *A. niger* N1 and O1, the first thing we identified was appropriate antibiotics for selection. *A. niger* N1 and O1 were cultivated on bottom-agar plates with hygromycin B, G418, or phosphinothricin at different concentrations. *A. niger* N1 was completely inhibited by 50 μg/mL hygromycin and 200 μg/mL G418 (Figure 2A). *A. niger* O1 was completely inhibited by 200 μg/mL hygromycin and 300 μg/mL G418 (Figure 2B). Both *A. niger* N1 and O1 grew normally at less than or equal to 200 μg/mL and 300 μg/mL phosphinothricin, respectively (Figure 2A,B). Taken together, both *A. niger* N1 and O1 showed sensitivity to hygromycin and G418, and they were insensitive to phosphinothricin (Figure 2A,B). Additionally, *A. niger* N1 appeared to be more sensitive to hygromycin and G418 than *A. niger* O1 (Figure 2A,B). 

We chose concentrations of antibiotics that completely inhibited the growth of *A. niger* N1 and O1 during the process of genetic transformation. Therefore, final concentrations of 50 μg/mL hygromycin B and 200 μg/mL G418 were used as screening concentrations during the genetic transformation of *A. niger* N1. Meanwhile, final concentrations of 200 μg/mL hygromycin B and 300 μg/mL G418 were used as screening concentrations during the genetic transformation of *A. niger* O1.

### 3.3. Preparation of Protoplasts of A. niger N1 and O1

To obtain a high transformation efficiency, it is essential to generate enough protoplasts. The quantity of protoplast production can be affected by several factors: the composition of the culture medium, the age of the mycelia used, the lytic enzymes used, and the enzymolysis time. To find the best medium to culture *A. niger* N1 for protoplast production, mycelia were cultured in five different media at 30 °C. The highest yield of *A. niger* N1 protoplasts (9.2 × 10^6^/mL) was obtained from CPZ-old medium; there was also little mycelial debris in this medium. The yields of the protoplasts decreased successively in media MM1, PDA, MM2, and EP. PDA and EP media released fewer protoplasts and significant mycelial debris (Figure 3A,B). Consequently, CPZ-old medium was the best among the tested media (Figure 3A,B). Compared to the media (MM1 (Vogel’s minimal medium) [52], MM2 (complete medium) [14], PDA, EP [53]) used in previous studies, the nutrients in CPZ-old medium were relatively simple, with fewer inorganic salts in terms of type.

The effect of mycelial age on protoplast yield was then investigated. With increasing culture time, the yield of protoplasts increased first and then decreased; it peaked at 9.2 × 10^6^/mL when the culture time was 21 h (Figure 3C). Additionally, the enzymatic combinations had a significant influence on the yield of protoplasts. It was found that 1% lysing enzymes from *T. harzianum* alone was best, producing a protoplast yield of 9.0 × 10^6^/mL. The protoplast yields were slightly decreased when other lytic enzymes were added (Table 1). Furthermore, the incubation time was an important factor for protoplast release because shortened or prolonged incubation with lytic enzymes resulted in the incomplete formation of protoplasts or the degradation of early formed protoplasts. The optimum enzymatic incubation period was determined by incubating mycelium for up to 5 h with cell wall lytic enzymes. The highest yield (11 × 10^6^/mL) was observed after 2 h of incubation (Figure 3D). Importantly, the enzymatic hydrolysate needed to be removed as soon as possible to avoid excessive protoplast incubation. In summary, the maximum protoplast production from *A. niger* N1 (11 × 10^6^/mL) was obtained from old mycelia that were 21 h old that had been cultured on CPZ-old medium and digested with 1% *T. harzianum* lysing enzymes and incubated at 30 °C for 2 h.

Three media—CPZ-old, MM1, and MM2—were tested for protoplast preparation from mycelium of aconidial *A. niger* O1. The highest yield of protoplasts (16.9 × 10^6^/mL) was obtained when using CPZ-old medium. Therefore, we chose CPZ-old as the culture medium for *A. niger* O1 (Figure 4A). We optimized the culture time of *A. niger* O1 and found that mycelium cultured for 19 h produced the highest yield of protoplasts; a steep decrease in the protoplast yield was observed when the culture time was changed to 12 or 26 h (Figure 4B). Interestingly, unlike for *A. niger* strain N1, few protoplasts were obtained from A. *niger* O1 using lysing enzymes from *T. harzianum* alone, and the addition of other lytic enzymes was necessary to obtain a sufficient yield of protoplasts. Up to 15 × 10^6^/mL protoplasts were obtained when using five lytic enzymes preparations together (lysing enzymes from *T. harzianum*, Lysozyme, Snailase, Driselase, and Yatalase); however, much of the mycelial debris could not be filtered out (Table 2). To decrease the amounts of mycelial debris, we removed each lytic enzyme preparation from the cocktail in turn. The mycelial debris disappeared after the removal of Yatalase, while the removal of any of the other lytic enzymes did not alter the amounts of mycelial debris. The yield of protoplasts was maximal (17 × 10^6^/mL) when the lytic enzyme combination used included 1.2% lysing enzymes from *T. harzianum*, 0.5% Lysozyme, and 0.5% Snailase (Table 2). The final results showed that the maximum production of protoplasts (17 × 10^6^/mL) from *A. niger* O1 was obtained from mycelia that were 19 h old that had been cultured on CPZ-old medium and digested with a multiple-lytic-enzyme combination including 1.2% lysing enzymes from *T. harzianum*, 0.5% Lysozyme, and 0.5% Snailase.

### 3.4. PMT and AMT of A. niger N1 and O1

We constructed the binary vector pAN52-P*ahr*-*gfp*-T*ahr* (Figure 5A) using the methodology described above (see Section 2.1). The GFP cassette fragment was amplified from this vector using the primers 1F/1R and was used for the transformation of *A. niger* N1 and O1 into protoplasts. Unlike in a previous method that used shaken liquid culture to obtain young mycelia [14,50], herein, young mycelia of *A. niger* N1 (which produces few conidia) and aconidial *A. niger* O1 generated on solid medium were used for the release of protoplasts. For *A. niger* N1, 10^7^ conidia were needed to obtain sufficient protoplasts using solid-medium plates, whereas in previous work, 10^8^ conidia were incubated in liquid culture to generate a sufficient yield of protoplasts [14,50]. *A. niger* strain O1 forms small colonies on agar plates, but it is difficult to produce dispersed mycelia in shaken-flask culture, while 3–6 days are needed to obtain mycelia via static culture in liquid. Thus, compared to static culture, the solid-medium method used here saved time. Therefore, the solid-medium culture method for the release of protoplasts that we applied has advantages for both *A. niger* N1 and O1.

Putatively transformed colonies were screened by PCR analysis. It was found that 25 of 28 transformants into *A. niger* N1 by PMT were correct, as identified using primer pairs GFP-1F/1R and NEO-1F/1R. There was no amplification of the target genes in the negative controls (Appendix A). For *A. niger* O1 and PMT, 32 of 39 transformants were correct (Appendix A). Ten PCR- and fluorescence-confirmed transformants were randomly selected to observe the stability of the fluorescence. All of the transformants had stable fluorescence signals determined by the methods described in Section 2.4 (Figure 6A,C). In total, 69 and 16 positive transformants in each transformation plate (there are seven plates in one transformation process) of *A. niger* N1 and O1 were obtained by PMT, respectively (Figure 6E). Taken together, the transformation efficiencies obtained by PMT were 89.3% (25/28) and 82.1% (32/39) for *A. niger* N1 and O1, respectively (Appendix A). 

Separately, pPK2-*hph*-*gfp* (Figure 5B) was transformed into conidia of *A. niger* N1 and young mycelia of *A. niger* O1 by AMT. The age of the mycelia of *A. niger* O1 was the vital factor for the success of the AMT method—the optimum culture time of *A. niger* O1 mycelia was 8 h, while the AMT method did not result in transformants when applied to *A. niger* O1 if the culture time was 15 h. Acetosyringone (AS) served as an inducer of virulence genes, the expression of which is a prerequisite for T-DNA transfer [24]. The AS concentration during the *Agrobacterium* co-cultivation period and the preculture period of *A. tumefaciens* influenced the frequency of transformation during the AMT of *S. cerevisiae* and several filamentous fungi [21]. A previous study using *M. thermophila* reported that the maximum transformation efficiency was obtained with 200 μM AS [21]. Therefore, the concentration of AS used in our experiments was 200 μM. Here, 55 of 56 AMT-mediated transformants of *A. niger* N1 and 28 of 64 transformants of *A. niger* O1 were found to be positive using the primers GFP-2F/2R and HPH-1F/1R (Appendix A). Additionally, 129 and 38 positive transformants in each transformation plate were obtained for *A. niger* N1 and O1 by AMT, respectively (Figure 6E). The transformation efficiencies by AMT were 98.2% (55/56) and 43.8% (28/64) for *A. niger* N1 and O1, respectively (Appendix A). 

To validate the integration of the transformed DNA into the host genome in more detail, we determined the copy number(s) of the integrated *gfp* gene in the positive transformants by real-time quantitative PCR and Southern blot analysis. Six PCR and fluorescence-confirmed PMT transformants of *A. niger* N1 with the *gfp* reporter gene (Appendix A: PMT-*gfp* 1–6) and six AMT transformants with the *gfp* reporter gene (Appendix A: AMT-*gfp* 7–12) were randomly selected. Similarly, six PMT transformants of *A. niger* O1 with the *gfp* reporter gene (Appendix A: PMT-*gfp* 1–6) and six AMT transformants with the *gfp* reporter gene (Appendix A: AMT-*gfp* 7–12) were randomly selected. The *gfp* gene of all the AMT transformants was present in a single copy in both *A. niger* N1 and O1. This is consistent with a previous study in *A. giganteus* [49]. Meanwhile, the *gfp* gene of most of the PMT transformants was present in multiple copies in both *A. niger* N1 and O1 (Appendix A).

To further confirm this observation, we randomly selected three and two PCR-confirmed transformants produced using PMT and AMT, respectively, and performed Southern blot analysis of them. Among the three transformants prepared using PMT, there was a single-copy *gfp* gene insertion, a two-copy insertion, and a three-copy insertion (Figure 6F). The two transformants obtained by AMT were single-copy insertions. Additionally, it seems that the integration of the introduced DNA is completely at random within the chromosome (Figure 6G). The probe did not yield bands with genomic DNA from untransformed *A. niger* N1 (Figure 6F,G).

### 3.5. CRISPR/Cas9 Genome Editing System and Marker-Free albA Gene Knockout Assay in A. niger N1

We constructed the vector pFC332-*AnalbA,* and the donor DNA fragment transformed into protoplasts of *A. niger* N1 (Figure 7A) as described above (see Section 2.5). To demonstrate the functionality of the CRISPR/Cas9 system in *A. niger* N1, we chose to target the *albA* gene (ANI_1_726084). Knockout of *albA* results in the formation of white-colored colonies, thus providing a direct indication on transformation plates. A series of transformations of pFC332-*A**nalbA* were performed in *A. niger* N1 (Appendix A). To verify whether the *albA* gene had been knocked-out, 22 randomly selected single transformation colonies were analyzed the PCR using primer pair *AnalbA*-DE-F/R (Appendix A). *A. niger* N1 (which has a Ku70 ortholog) can repair a Cas9-induced double-strand break through nonhomologous end joining (NHEJ). However, as shown in Appendix A, the transformation of *A. niger* N1 with pFC332-*AnalbA* yielded about five colonies in each transformation plate (total seven plates), but the *AnalbA* gene was not knocked out in 22 randomly selected colonies without a homology template (Appendix A). Transformation with the control plasmid, pFC332, which lacks an U6p-*AnalbA* Target-sgRNA expression cassette, resulted in 40 viable colonies (Appendix A), and, as expected, PCR showed that there were no *AnalbA*-knockout transformants among them (Appendix A).

Co-transformation of a homology repair DNA fragment consisting of fused 5′- and 3′- flanking regions of the *AnalbA* gene, together with pFC332-*AnalbA*, yielded 9 (out of 22) white transformants on the transformation plates (Appendix A). Streaking of one such transformant on a CPZ-new agar plate containing hygromycin showed the white coloration more clearly (Appendix A). A control co-transformation of pFC332 without the sgRNA targeting plasmid and the homology repair DNA fragment did not yield knockout colonies (Appendix A), showing that the repair DNA fragment does not integrate autonomously at the site of homology without the assistance of the Cas9–sgRNA targeting plasmid. A streak of one such transformant is shown in Appendix A; the difference in color compared to the *AnalbA* knockout shown in Appendix A is clear.

White transformants produced by the co-transformation of the homology repair DNA fragment and pFC332-*AnalbA* were cultured on CPZ-new plates without hygromycin as described in “Methods 2.5” to assess whether the transformants could lose the pFC332-*AnalbA* plasmid in the absence of selective pressure. The nine *AnalbA*-knockout transformants then failed to grow in medium containing 50 μg/mL hygromycin, thus indicating that they had lost the plasmid (Figure 7B). They were further analyzed by PCR using the primers HPH-DE-F/R to determine if the hygromycin resistance gene had been lost. There was no amplification of the hygromycin resistance gene from the nine *AnalbA*-knockout transformants or the *A. niger* N1 negative control, but a 1006-bp fragment was amplified from the positive control plasmid (Appendix A). Thus, we demonstrated that the *AnalbA*-knockout transformants could lose the plasmid (Figure 7B) that enables the construction of marker-free gene knockouts.

## 4. Discussion

The application of fungal protoplasts has contributed to the development of microbial strains and the genetic recombination of fungi. Because of the limitations of traditional mutation screening methods [61], improvement in the genetic transformation methods for industrial strains of filamentous fungi, especially aconidial *A. niger* O1, is urgently needed. In this study, we improved PMT and AMT techniques for *A. niger* N1 (which produces few conidia) and *A. niger* O1 (which is aconidial).

The physiological stage of the mycelium plays a significant role in the release of protoplasts [62,63]. Here, the culture medium used had a great influence on the growth of *A. niger*. Dispersed mycelium was observed when *A. niger* N1 was cultivated on CPZ-old agar plates with sucrose as the sole available extracellular carbon source (along with several inorganic salts), while the mycelia appeared densely when *A. niger* N1 was cultivated on MM1, MM2, EP, or PDA plates (Figure 1B). MM1 and MM2 contain a variety of trace elements, a single carbon source, and several inorganic ions. EP medium contains many trace elements, two carbon sources, and several inorganic ions. The nutrient composition of PDA is even more complex. These observations suggest that the dense mycelia cultured in nutrient-rich media were not conducive to enzymolysis and the release of protoplasts.

Mycelium age is an important factor for protoplast release, and different filamentous fungi behave differently [22,62,64]. With regard to the culturing time of *A. niger* N1, mycelia harvested at 21 h produced the highest yield of protoplasts, and a steep decrease in the protoplast yield was observed with the use of 15 or 27 h-old mycelia. Previous studies have reported that young and exponentially growing mycelia favored the release of protoplasts [22,62,64]. For instance, *A. niger* conidia were cultured in 100 mL complete medium (CM) for 12 h (at 100 rpm) to generate protoplasts [14]. It seems that young mycelia are more susceptible to enzymatic digestion than old mycelia and that the cell wall of old mycelia becomes thicker [22,63,64,65].

As previously reported, the biggest influencing factor on the release of protoplasts from mycelia was the choice of the enzymatic combination and the concentrations used [61,62]. The main components of fungal cell walls are polysaccharides, proteins, and lipids. The cell wall polysaccharides of *A. niger* mainly include chitin, cellulose, and glucan [66]. Taking *N. crassa* as an example, the outermost layer is an amorphous glucan, followed by a coarse web of glycoproteins, and the innermost layer is chitin [66]. According to the information provided by the manufacturers, the main components of the lysing enzymes from *T. harzianum* are cellulase, protease, chitinase, and glucanase. Because this preparation contains most of the lytic enzyme activities required to lyse the main components of the cell wall of filamentous fungi, it has been the most commonly used lytic enzyme mixture. However, other lytic enzymes may also need to be added for different filamentous fungi. Lysozyme mainly lyses peptidoglycan; Snailase mainly contains cellulase, protease, and pectinase; Driselase mainly contains laminarin, xylanase, and cellulase; and Yatalase mainly demonstrates chitinase and chitobiase activities. The lysing enzymes from *T. harzianum* alone were enough for the release of *A. niger* N1 protoplasts (Table 1). However, Lysozyme and Snailase were also needed for the release of protoplasts from *A. niger* O1 (Table 2). We speculate that the compositions of the cell walls of *A. niger* N1 and O1 are different even though both strains are descended from the same ancestor.

PMT has been generally used in many fungi, and the transformation efficiencies vary greatly in different fungi for different purposes [49,67], such as in *A. giganteus*, where there were 55 transformants in 10^8^ protoplasts/μg DNA obtained by PMT [49]. In this study, the PMT technique yielded up to 10 and 8 transformants in 10^7^ protoplasts per microgram of DNA in *A. niger* N1 and O1, respectively. Meyer et al. reported that the deletion of the Ku70 homologue from *A. niger* (KusA) dramatically improved the homologous integration efficiency of *AngfaA*, which reached >95% compared to 19% in the wild-type background when 1000-bp homologous flanks were used [67]. For the AMT system in filamentous fungi, previous studies reported that the transformation efficiencies varied greatly in fungal species: 0.04% for *Blastomyces dermatitidis* [68], 29% for *A. awamori* [69], 58% for *M. thermophila* [21], 74% for *Fusarium avenaceum* [70], and 85% for *F. graminearum* [70]. However, a previous study reported that the transformation frequency obtained via AMT of *A. niger* was not as high as expected and was not reproducible for unknown reasons [25]. Herein, the transformation efficiencies by AMT were 98.2% (55/56) and 43.8% (28/64) for *A. niger* N1 and O1, respectively (Appendix A).

While this manuscript was in revision, two papers regarding the strain engineering of *A. niger* N1 [71] and O1 [72] using the protoplast transformation procedure described in *M. thermophila* were published. In *A. niger* O1, Guo et al. described that the four overexpression and four knockout transformants of the acid α-amylase gene were simply obtained after protoplast transformation, while four overexpression and two knockout transformants of the neutral α-amylase gene were generated [72]. In *A. niger* N1, 6 out of 52 transformants were triple-gene mutants, with a gene editing efficiency of 11.5% [71]. Compared to previous studies [71,72], the PMT method we developed here is relatively simple. First, in previous studies, the young mycelia were washed twice with 1 mol/L MgSO_4_ before lysing with enzymatic hydrolysate [71,72], while in our study, enough protoplasts could be obtained without carrying out these washing steps; therefore, the PMT method that we developed is relatively simple and saves time. Second, previous studies washed the protoplasts with STC solution twice after the young mycelia had been lysed with enzymatic hydrolysate [71,72]; however, the protoplasts were only washed with solution B once in our study, so the method that we developed can shorten the time required for the protoplasts to remain in the osmotic stabilizer solution, thus reducing the damage to the protoplasts obtained by centrifuging twice. Third, there are some differences between the PMT method that we established in this manuscript and the method used in previous studies. For example, in the transformation procedure, the protoplast mixture was heated at 60 °C for 2 min after the DNA mixture was added into the protoplast STC solution [71,72]; however, no heat treatment is used in our method. Finally, although the flow cytometry-based plating-free system developed by Yang et al. [71] was more convenient than the PMT method used in our manuscript, the selection marker gene was still contained in the transformants generated by Yang et al., while an efficient marker-free gene deletion system was constructed in this manuscript. To increase the efficiency of PMT, we also tried using plasmids and linear DNA in the transformation; we found that the transformation efficiency was far lower for plasmid DNA than for linear DNA. Previous reports have stated that a larger plasmid size results in decreased transformation efficiency [73,74], indicating that it is harder for larger plasmid DNA to transform into cells.

The mode and frequency of individual integration events resulting from homology relate to both the transformation the host itself and the transformation technique [20]. Using AMT, single-copy integration events were detected in many filamentous fungi, including in *Verticillium fungicola*, *A. giganteus,* and *A. awamori* [24,49,75], whereas multiple integration events were frequently observed in the transformants obtained by PMT [49]. The assay of the copy numbers of the integrated *gfp* gene in Figure 6G and Appendix A showed that here, AMT resulted in single-copy integration events. Meanwhile, both single and multiple integrations occurred in PMT (in Figure 6F and Appendix A), consistent with previous works [49,75]. Two pathways—the homologous recombination (HR) and NHEJ pathways—have been described as mediating double-strand break repair [20]. HR requires the presence of homologous fragments and thus leads to targeted integration; in contrast, the NHEJ-mediated ligation of DNA strands results in random integration. Very low frequencies of HR result in gene targeting being difficult in filamentous fungi [20]. Therefore, it is likely that the exogenous GFP cassette fragment was randomly integrated into the genomes of *A. niger* N1 and O1 (which has a Ku70 ortholog) via NHEJ. Meanwhile, a double-strand break generated by the CRISPR/Cas9 system can be repaired by NHEJ to directly carry out indel mutagenesis; alternatively, homology-directed repair (HDR) can be accomplished with a DNA repair template (donor DNA) [44]. Therefore, the deletion of *AnalbA* using a marker-free CRISPR/Cas9 system in *A. niger N1* strain can be achieved by HDR if *AnalbA* donor DNA is provided with the sgRNA simultaneously.

## 5. Conclusions

Here, we targeted two industrial *A. niger* glucoamylase-producing strains, N1 (which produces few conidia) and O1 (which lacks conidia), for which the existing genetic manipulation methods are laborious. We provided high-efficiency genetic tools via the systematic optimization of the whole process. The method of culturing *A. niger* N1 and O1 mycelia using solid-medium plates rather than liquid medium for the release of protoplasts provides a case study for manipulating filamentous fungi with few or even no conidia. Following our systematic optimization, the transformation efficiency reached 89.3% for *A. niger* N1 and 82.1% for A. *niger* O1 by PMT; when using AMT, it was 98.2% for A. *niger* N1 and 43.8% for A. *niger* O1. These are good numbers of positive transformants, especially when considering that the transformation efficiency of A. *niger* by AMT is normally low [25]. We also developed a marker-free CRISPR/Cas9 genome editing system using an AMA1-based autonomously replicating plasmid to express the Cas9 protein, sgRNA, and the selectable marker *hph* gene for hygromycin. By using this technology, we constructed a marker-free *albA* deletion mutant that was visible due to its white coloration in the *A. niger* N1 strain. Our results indicate that the approaches that we improved are efficient for the genetic manipulation of *A. niger* N1 and O1 and will contribute to the use of efficient targeted mutation in other industrial strains of *A. niger*.

## Figures and Tables

**Figure 1 biology-11-01396-f001:**
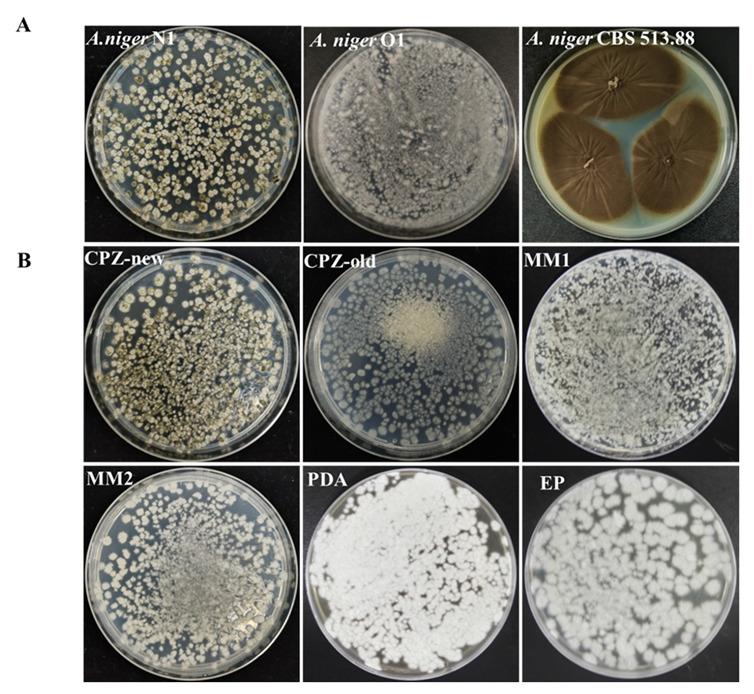
Growth phenotypes of *A. niger* N1, *A. niger* O1, and *A. niger* CBS513.88. (**A**) Conidiation of *A. niger* N1 after 7-day culture on CPZ-new medium in 9 cm Petri dishes (left image), aconidial *A. niger* O1 generated on CPZ-old medium for 7 days in 9 cm Petri dishes (middle image), and conidiation of *A. niger* CBS513.88 after 7-day culture on PDA medium in 9 cm Petri dishes (right image). (**B**) Conidiation of *A. niger* N1 after 7-day culture on CPZ-new medium in 9 cm Petri dishes (left image) and mycelia of *A. niger* N1 after 7 days of culture on several commonly used media (CPZ-new, CPZ-old, MM1, MM2, PDA, and EP).

**Figure 2 biology-11-01396-f002:**
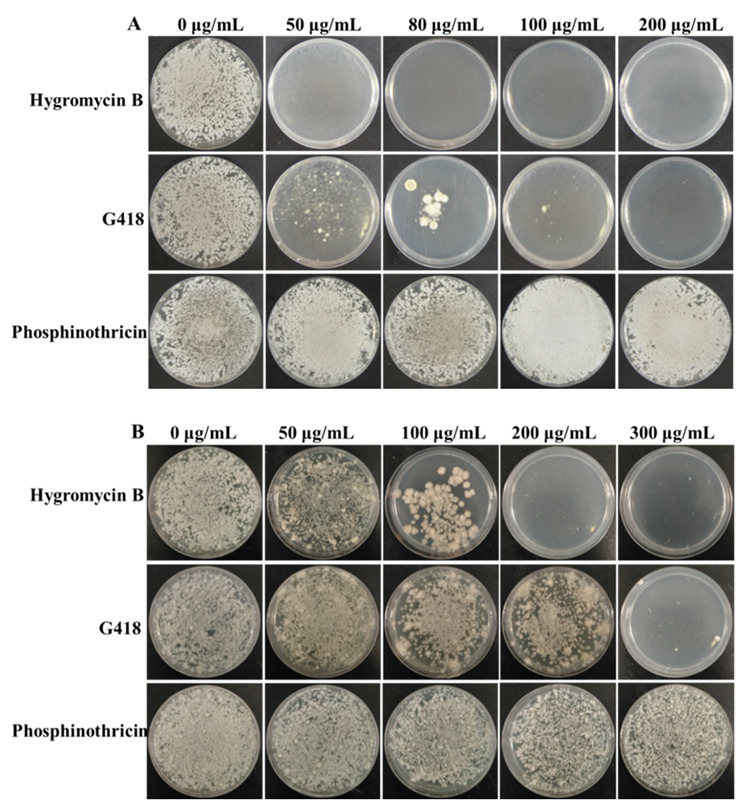
Marker screening for *A. niger* N1 and *A. niger* O1. (**A**) Five concentration gradients (0, 50, 80, 100, and 200 μg/mL) of three antibiotics (hygromycin B, G418, and phosphinothricin) were tested on bottom-agar plates at 30 °C for A. niger N1. (**B**) Five concentration gradients (0, 50, 100, 200, and 300 μg/mL) of three antibiotics (hygromycin B, G418, and phosphinothricin) were tested on bottom-agar plates at 34 °C for *A. niger* O1.

**Figure 3 biology-11-01396-f003:**
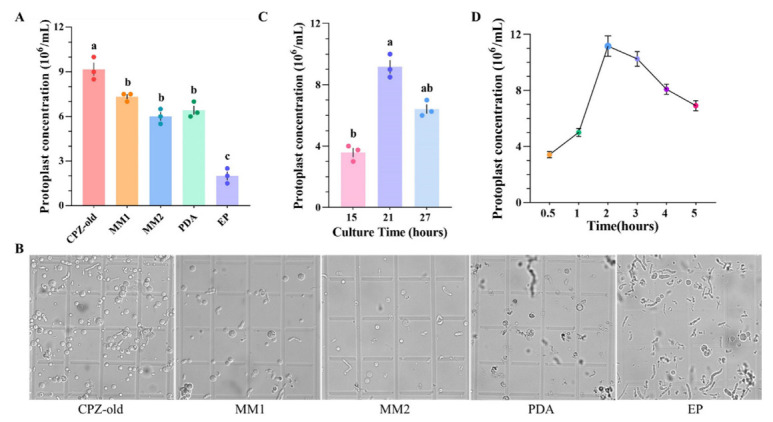
Optimization of protoplast production from *A. niger* N1 for protoplast-mediated transformation (PMT). (**A**) Column diagram of the numbers of protoplast production from *A. niger* N1 in different media (CPZ-old, MM1, MM2, PDA, and EP). (**B**) The protoplast production in each 4 × 4 square corner area of the hemocytometer for *A. niger* N1 in different media (CPZ-old, MM1, MM2, PDA, and EP). (**C**) Protoplast production at different culture times (15, 21, and 27 h) of *A. niger* N1 in CPZ-old medium. (**D**) Protoplast production with different lysing times (0.5, 1, 2, 3, 4, and 5 h) of *A. niger* N1 cultured on CPZ-old medium. Data represent mean ± SD. Means with different lowercase letters (a–c) in each graph differed significantly (Tukey’s HSD, *p* < 0.05). Error bars: SD of the mean from three repeated assays.

**Figure 4 biology-11-01396-f004:**
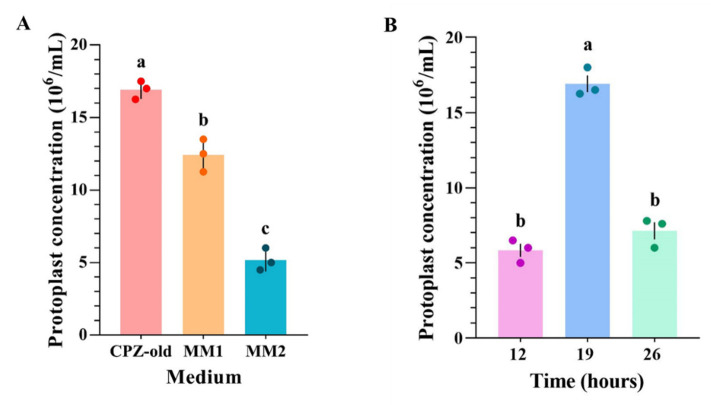
Optimization of protoplast production for PMT of *A. niger* O1. (**A**) Column diagram of the numbers of protoplast production from *A. niger* O1 in different media (CPZ-old, MM1, and MM2). (**B**) Protoplast production from *A. niger* O1 with different culture times (12, 19, and 26 h) in CPZ-old medium. Data represent the mean ± SD. Means with different lowercase letters (a–c) in each graph differed significantly (Tukey’s HSD, *p* < 0.05). Error bars: SD of the mean from three repeated assays.

**Figure 5 biology-11-01396-f005:**
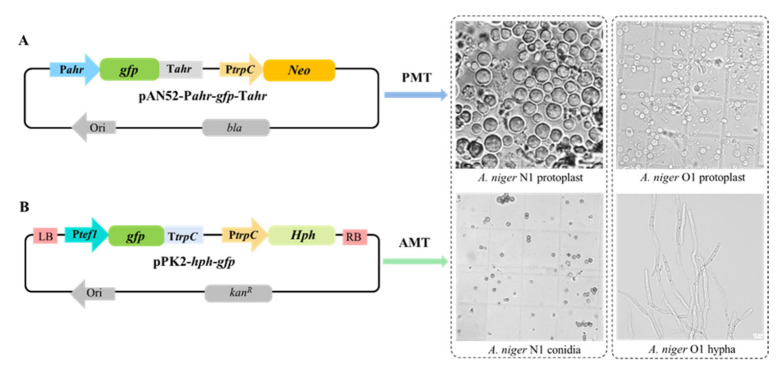
Transformation of *A. niger* N1 and O1 was carried out by protoplast-mediated transformation (PMT) using the vector of pAN52-P*ahr*-*gfp*-T*ahr* (**A**) and *Agrobacterium tumefaciens*-mediated transformation (AMT) using the vector of pPK2-*hph*-*gfp* (**B**). pAN52-P*ahr*-*gfp*-T*ahr* was transformed into protoplasts of *A. niger* N1 and O1 by PMT. The pPK2-*hph-gfp* was transformed into conidia of *A. niger* N1 and young mycelia of *A. niger* O1 by AMT. P*ahr* is the promoter of the alkyl hydroperoxide reductase from *M. thermophila*; P*tef1* is the promoter of the translation elongation factor gene *tef1* from *A. niger*; P*trpC* is the promoter of the tryptophan synthetase gene from *A. nidulans*; GFP, enhanced green fluorescence protein; *Neo*, neomycin-resistance gene; *Hph*, hygromycin resistance gene; RB and LB, right and left border of T-DNA, respectively.

**Figure 6 biology-11-01396-f006:**
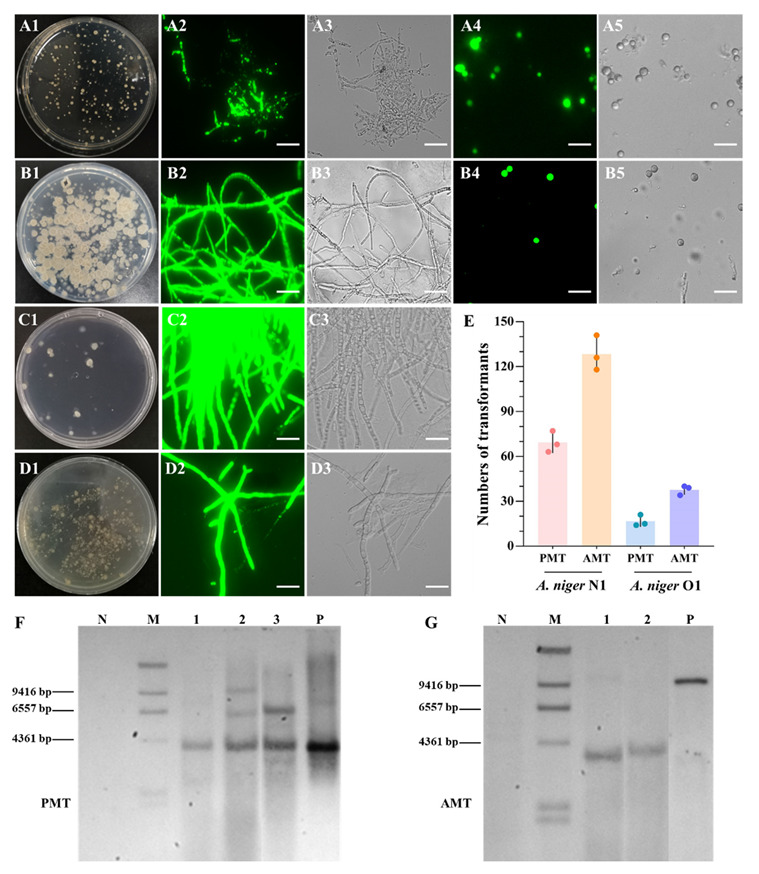
Confirmation of *A. niger* N1 and O1 transformants. (**A**) *A. niger* N1 PMT transformation plate (**A1**), mycelial fluorescence (**A2**), mycelia in bright field (**A3**), conidial fluorescence (**A4**), and conidia in bright field (**A5**). (**B**) *A. niger* N1 AMT transformation plate (**B1**), mycelial fluorescence (**B2**), mycelia in bright field (**B3**), conidial fluorescence (**B4**), and conidia in bright field (**B5**). (**C**) *A. niger* O1 PMT transformation plate (**C1**), mycelial fluorescence (**C2**), and mycelia in bright field (**C3**). (**D**) *A. niger* O1 AMT transformation plate (**D1**), mycelial fluorescence (**D2**), and mycelia in bright field (**D3**). (**E**) Numbers of transformants with fluorescence in each transformation plate. (**F**) Southern blot analysis of *A. niger* N1 PMT transformants. P, the *gfp*–*neo* cassette fragment digested with *Xho*I as a positive control (expected product of 3908 bp); lanes 1–3, transformants obtained by PMT. (**G**) Southern blot analysis of *A. niger* N1 AMT transformants. P, the pPK2-*hph*-*gfp* plasmid digested with *EcoR*V as a positive control (expected product of 10,209 bp); lanes 1–2, transformants obtained by AMT. N, genomic DNA from untransformed *A. niger* N1 as a negative control; M, DNA Molecular-Weight Marker II. Scale bars in panels A–D, 20 µm.

**Figure 7 biology-11-01396-f007:**
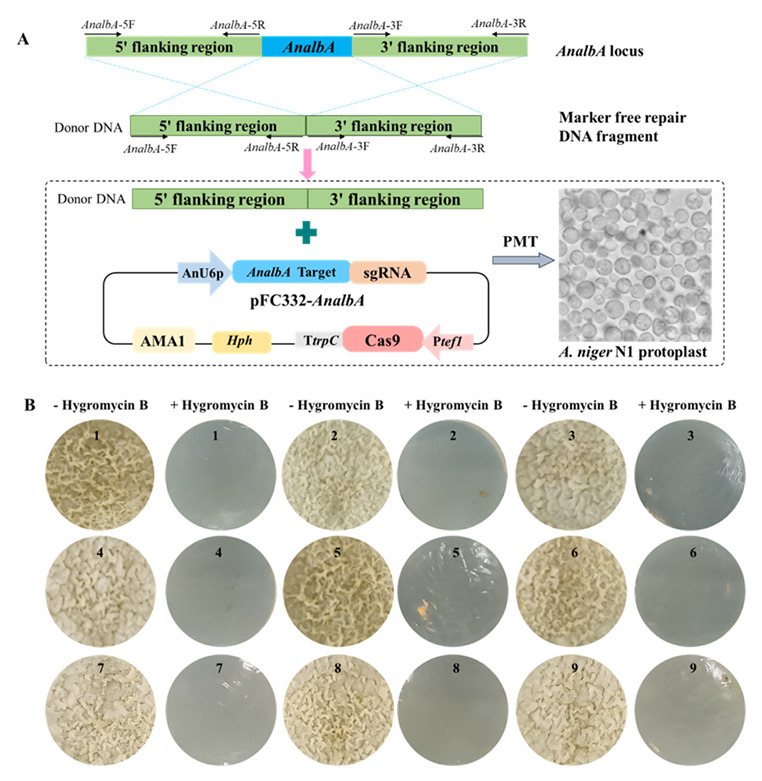
Generation of the marker-free *AnalbA*-knockout mutants of *A. niger* N1 by autonomously replicating vector harboring Cas9- and sgRNA-encoding genes and a hygromycin sensitivity test of *AnalbA*-knockout mutants before and after plasmid removal. (**A**) pFC332-*AnalbA* and the donor DNA fragment transformed into *A. niger* N1 by PMT. AnU6p is the promoter of *A. niger* N1; Ptef1 is the promoter of translation elongation factor gene *tef1* from *A. nidulans*; the *cas9* gene encoding Streptococcus pyogenes Cas9 was codon-optimized for expression in *A. niger*; AMA1, self-replicating extrachromosomal AMA1 elements from *A. nidulans*; *Hph*, hygromycin resistance gene. (**B**) Phenotypes of nine *AnalbA*-knockout transformants with loss of pFC332-*AnalbA* plasmid with and without hygromycin pressure. Mutant with *AnalbA* knockout and lost pFC332-*AnalbA* plasmid failed to grow in medium containing 50 μg/mL hygromycin.

**Table 1 biology-11-01396-t001:** Effect of different enzymatic combinations on the release of *A. niger* N1 protoplasts.

Lysing Enzymes from *T. harzianum* (%, *w/v*)	Lysozyme (%, *w/v*)	Snailase (%, *w/v*)	Driselase (%, *w/v*)	Number of Protoplasts (10^6^/mL)
1	0	0	0	9.0
1	0.5	0	0	7.5
1	0	0.2	0	8.25
1	0.5	0.2	0	8.75
1	0.5	0.2	0.2	8.25
1.5	0.5	0.2	0.2	8.5

**Table 2 biology-11-01396-t002:** Effect of different enzymatic combinations on the release of *A. niger* O1 protoplasts.

Lysing Ernzymes fom *T. harzianum* (%, *w/v*)	Lysozyme	Snailase	Driselase	Yatalase	Number of Protoplasts
	(%, *w/v*)	(%, *w/v*)	(%, *w/v*)	(%, *w/v*)	(10^6^/mL)
1	0	0	0	0	0.05
2	0	0	0	0	0.05
1.5	0.5	0.5	0.5	0.5	15 (with mycelial debris)
1.5	0.5	0.5	0.5	0	12 (no mycelial debris)
1.5	0.5	0.5	0	0.5	12 (with mycelial debris)
1.5	0.5	0.5	0	0	15 (no mycelial debris)
1.2	0.5	0.5	0	0	17 (no mycelial debris)
0.5	1.2	0.5	0	0	9 (no mycelial debris)
0.5	0.5	1.2	0	0	5 (no mycelial debris)
1.2	0.5	0	0	0	6 (no mycelial debris)
1.2	0	0.5	0	0	4 (no mycelial debris)

## Data Availability

All data used to support the findings of this study are included within the article and they are also available from the corresponding author upon request.

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
