# Peer review of "Development of Genetic Tools in Glucoamylase-Hyperproducing Industrial Aspergillus niger Strains"

_biology, 2022, doi:10.3390/biology11101396_

Round 1
Reviewer 1 Report
In this manuscript, the authors successfully developed high-efficiency transformation tools including protoplast-mediated transformation and Agrobacterium tumefaciens-mediated transformation for the genetic manipulation of two industrial glucoamylase-producing strains A. niger N1 and O1. Moreover, the authors developed a marker-free CRISPR/Cas9 genome editing system using an AMA1-based plasmid to express Cas9 protein, sgRNA, and the selectable marker. By using this technology, marker-free albA deletion mutants were generated in the A. niger N1 background. Overall, this manuscript provides a valuable reference for facilitating strain engineering of the industrially valuable A. niger strains. However, some points need to be addressed prior to publication as detailed below.
1. The figures and figure legends should be more informative and accurate.
Line 322-323: The information in Figure 1A does not correspond to the figure legend.
Line 495: The legend in Figure 5 does not provide a complete description of the information in Figure 5, only information on vector maps is included and information on the transformation strategy is missing.
Line 556: The phrase "Components of the marker-free CRISPR–Cas9 system used to target the AnalbA gene." in the Figure 7 legend cannot summarize all the information in Figure 7.
2. Line311: According to Fig. 1B, A. niger N1 produced conidia on the CPZ-new plate. However, the authors describe the result as "The mycelia of A. niger N1 grew well on various commonly used media, but no conidia were produced (Figure 1B)", which needs to be expressed more accurately.
3. According to the results of this manuscript, the AMT method had a higher transformation efficiency compared to the PMT method. Why was the PMT method chosen to construct the marker-free CRISPR/Cas9 system rather than the AMT method?
4. According to the data in Table 1, the addition of Snailase resulted in an increased concentration of protoplasts released from the enzymatic combination containing Lysing enzymes from T. harzianum and Lysozyme, suggesting that Snailase may contribute to protoplast release. Did the authors explore the protoplast concentration of enzymatic combination containing Lysing enzymes from T. harzianum and Snailase?
5. Line 487-488: "Among the three transformants prepared using PMT, there was a single-copy T-DNA insertion, a two-copy insertion, and a three-copy insertion." The PMT method did not use T-DNA, so why T-DNA copy numbers were detected?
6. In the discussion section, the authors only give the results of two papers on the engineering of A. niger N1 [66] and O1 [67] strains as references but do not compare the advantages and disadvantages of the methods developed in this manuscript with those used in previous studies.
Minor comments
1. "the" should be added before "genetic manipulation" in Line 67, "marker-free" in Line 125, and "homologous integration" in Line 450, etc.
2. Line 83: "had" should be replaced by "has"
3. Line 102: " level" should be replaced by " levels" ("level" should be plural).
4. Line 204: "was" should be added before "poured", and "medium" should to be replaced by " media" ("medium" should be plural).
5. Line 271: "A" should be lowercase.
6. Line 335: the comma before "was" should be deleted.
7. Line 357: "and" should be added before "the enzymolysis time".
8. The number of decimal places retained in the percentage data should be the same (Line 443, 467, etc.).
9. Line 522: "was" should to be replaced by "were".
10. Line 591: "mycelium" should be plural.
11. Line 620: "homologous" should be replaced by "homology".
12. Line 631: "resulted" should be replaced by "results".
etc.
Reviewer 2 Report
In this paper, the authors explored successfully high-efficiency transformation tools, including protoplast-mediated transformation and Agrobacterium tumefaciens-mediated transformation, as well as marker-free CRISPR/Cas9 genome editing system, in the glucoamylase-producing industrial strains A. niger N1 and O1, which is very helpful for molecular breeding for improving glucoamylase production. This manuscript is supported for acceptation for publication, but some comments are listed below.
1. In section 3.1, the production of conidia by A. niger is difficultly determined through direct investigation of colony phenotypes on solid plates in Fig. 1. The quantitative analysis of conidia is suggested to be performed.
2. The detail figure legends should be addressed, especially in Figs. 1 and 2.
3. In Fig. 3C and D, Fig. 4B, horizontal coordinate should not show ‘h’.
4. Fig. 3B, the pictures are blurry, resulting in the unclear protoplast.
Reviewer 3 Report
The manuscript of Liu et al reports on an improved method for genetic engineering of industrial-relevant A. niger strains. This is a timely and important study, as for fungi genetic tools for efficient and straightforward genetic manipulation are still scarce. The manuscript is clearly written, and described experiments appear to have been carefully conducted. I enjoyed reading the manuscript and apart from minor comments recommend publication.
Introduction: To emphasize the importance of their study, the authors may include numbers on the transformation efficiency or general shortcomings regarding established methods. Where is the benchmark to which they compare their own findings?
134ff: Please state why A. niger CBS513.88 was included in the study. This strain is not introduced in the manuscript and also does not appear in any following experiment as far as I understood.
Results 3.1: What was the intention of the growth / phenotype evaluation? Why is this important for method development?
Results 3.2: This part reads very difficult. Please shorten it significantly. Relevant data, which are apparent from the figure are not necessary to all repeat in the text. Just put highlights / key conclusions there and don’t comment on every single AB concentration.
L363: I did not find any data regarding the medium composition in reference 53. Please check, if this reference is correct here.
L444 – 451 and 468 - 473: Belongs to discussion
Figure 6: I suggest working with numbers (A1, A2,....) instead of “leftmost”, “middle”, etc.
Round 2
Reviewer 2 Report
Accept in present form